# Effect of Modified Silica Fume Using MPTMS for the Enhanced EPDM Foam Insulation

**DOI:** 10.3390/polym13172996

**Published:** 2021-09-03

**Authors:** Rudeerat Suntako

**Affiliations:** Department of Physics, Faculty of Liberal Arts and Science, Kasetsart University Kamphaeng Saen Campus, Nakhon Pathom 73140, Thailand; faasrrs@ku.ac.th

**Keywords:** silica fume, EPDM, MPTMS, filler, rubber, surface modification

## Abstract

Silica fume (SF) is a by-product from the production of silicon metal, which has a relatively high silica concentration. The surface modified silica fume (mSF) is treated with (3-mercaptopropyl) trimethoxysilane (MPTMS) as filler in ethylene propylene diene monomer (EPDM) foam. The FTIR spectra of mSF clearly indicated that MPTMS can be successfully bonded to the SF surface. The reinforcing efficiency of mSF-filled EPDM foam insulation indicated that the mechanical properties such as hardness, tensile strength, modulus, and compression set enhanced higher than in case of SF and calcium carbonate. While the cure characteristics such as the maximum torque (M_H_), the minimum torque (M_L_) and the differential torque (M_H_-M_L_) are increasing in proportion to increasing filler contents, mainly with mSF. For the cure behavior, the mSF-filled EPDM foam insulation showed the fastest cure time (t_c90_) and scorch time (t_s2_) due to reduced accelerator adsorption. Whereas, the calcium carbonate-filled EPDM foam insulation increased the cure time (t_c90_) and scorch time (t_s2_), therefore, it also prevents compound scorching. The results indicated that the mSF with MPTMS can be used as an alternative filler for EPDM foam insulation.

## 1. Introduction

EPDM foam insulation is a flexible and low density and light weight product. This product is widely used to save energy and prevent condensation problems with chilled water and refrigeration systems. It is always made from ethylene propylene diene monomer (EPDM) rubber because this provides good resistance to ozone, heat, and weather [1,2,3,4]. However, EPDM foam rubber has to fill the filler to improve the mechanical properties of rubber, especially tensile strength. In general, the carbon black [5,6] and silica [7,8,9,10,11,12,13] are a good filler that are always used to increase the mechanical properties of rubber. Nevertheless, it causes not only high costs but also environmental pollution problems in the manufacturing process [14,15]. Rubber scientists began using mineral fillers such as calcium carbonate [16,17], talc, and clays [18,19] because of their low cost [18] and eco-friendly despite its low strength (semi-reinforcing type). Therefore, rubber scientists have attempted to find a new filler that is low in cost, while offering the ability to reinforce rubber mechanical properties as well as being environmentally friendly. Silica fume (SF) is a by-product that fumes collected from the production of silicon metal. It contains more than 90% of silicon dioxide (SiO_2_) and is used in various applications, such as concrete, grouts, mortars, and fiber cement. Silica fume is a very fine powder consisting of spherical particles with a high specific surface area. Additionally, silica fume (SF) contains silanol groups on the surface. This is the same characteristic of silica but particle sizes are in the range of microns. Therefore, organosilane coupling agents such as (3-mercaptopropyl) trimethoxysilane (MPTMS) [10,20], bis-(3-triethoxysilylpropyl) tetrasulfane (Si-69) [21,22,23,24,25,26], 3-thiocyanatopropyl triethoxy silane (Si-264) [25], Cetyltrimethyl ammonium bromide (CTAB) [22] and 3-octanoylthio-1-propyltriethoxysilane (NXT) [27] are used to modify the surface of filler which is improve its reinforcing as well as processing characteristics. Therefore, it offers promising potential as an alternative filler for EPDM foam insulation. This study focuses on its rubber mechanical properties, which have not been reported elsewhere.

In this work, the silica fume is modified the surface using (3-mercaptopropyl) tri-methoxysilane (MPTMS) as a silane coupling agent because the latter acts as a bonding agent between EPDM rubber and silica fume to enhance reinforcing efficiency. Fourier transform infrared spectroscopy (FTIR) was used to analyze the surface modification of silica fume by MPTMS. The purpose of this work was to study the effects of the addition of silica fume (SF), modified silica fume with MPTMS (mSF), and calcium carbonate on the cure characteristics and the mechanical properties of EPDM foam insulation. At the same time, the structure and the compression set of EPDM foam insulation, which is filled by SF, mSF, and calcium carbonate, was also carried out.

## 2. Materials and Methods

### 2.1. Materials

The EPDM rubber (Keltan 8550C, ML1+4 (125 °C) 80 MU, ENB 5.5%, Ethylene content 48%) was purchased from Arlanxeo, Maastricht, Netherlands. ZnO was white seal grade and obtained from the Thai-Lysaght, Thailand. Carbon black was N-550 grade and manufactured by Thai Tokai Carbon Product, Thailand. Calcium carbonate was Omyacarb 1T and manufactured by Surint Omya Chemicals, Thailand. Process oil (PS-430T) was obtained from Idemitsu Lubricant, Thailand. Whereas stearic acid and sulphur were purchased from Chemmin, Thailand. The EM-80MA (OBSH grade) was used as a blowing agent and supplied by Eiwa Chemical, Thailand. Tetramethylthiuram disulfide (TMTD), zinc dimethyl dithiocarbamate (ZDMC), 2,2-dithiobis-(benzothiazole) (MBTS) was used as an accelerator and supplied by Kawaguchi Chemical Industry, Tokyo, Japan. Silane coupling agent was (3-mercaptopropyl) trimethoxysilane (MPTMS) and supplied by JJ-Degussa, Bangkok, Thailand. The silica fume (SF); the chemical compositions mostly consisted of silicon dioxide (about 95%). The average primary size was of 0.25 μm and the specific surface area was 20 m^2^/g.

### 2.2. Modification of Silica Fume Surface

The mSF modified the SF surface by (3-mercaptopropyl) trimethoxysilane (MPTMS). The 7 g of MPTMS was added to the 100 mL of ethanol and then stirring for 30 min. Then, 100 g of SF was added to the solution and then stirred for 15 min. After that, the mSF was dried at 100 °C for 12 h. The surface treatment was investigated by Fourier transform infrared spectroscopy (FTIR).

### 2.3. Compounding and Vulcanization

The formulation of EPDM foam is showed in Table 1. Raw materials were mixed in an internal mixer 3Lwith a rotor speed of 35 rpm. The Keltan 8550C, ZnO, stearic acid, PS-430T, N-550, omyacarb 1T or silica fume were added to the mixer, then mixed for 4 min, then ram up and mixed again for 3 min. The compound was eventually dumped and then TMTD, ZDMC, MBTS, OBSH, and sulphur were added on a two-roll mill at room temperature, then mixed for 3 min. Finally, the specimens were prepared by compression molding. The first step (prevulcanization) involved curing the compound in a mold at 150 °C for 5 min and the second step curing the compound in a hot air oven at 200 °C for 15 min.

### 2.4. Cure Characteristics and Mechanical Properties

The cure characteristics of the compound were investigated, according to ISO 6502 at 170 °C for 6 min, using with a Moving Die Rheometer (MDR2000, Alpha Technologies). The mechanical properties, such as hardness, tensile strength, 100% and 300% modulus, elongation at break and specific gravity were also investigated. A tensile testing machine (AG-IS, Shimadzu) was used for determining tensile strength, 100% and 300% modulus and elongation at break of the samples at 23 ± 2 °C, with an extension speed of 500 mm/min, according to ASTM D412. The specific gravity was investigated from the ratio of the density of EPDM foam to the density of water, according to ASTM D297. The hardness test was performed using a hardness tester (Teclock), according to ASTM D2240 with a Shore OO durometer.

### 2.5. EPDM Foam Structures and Compression Set

The cell structures of EPDM foam insulation were characterized with a digital microscope at 100X (VHX-500F-Lens 100X, Keyence). The specimens were cut from the EPDM foam insulation. The cell size of the EPDM foam insulation in the foam matrix was observed. The compression set was investigated according to ASTM D395, and the specimens were compressed to 75% of their original height for 48 h at 100 °C.

## 3. Results and Discussion

### 3.1. Characterization of the Modified Silica Fume (mSF)

Figure 1 shows the FTIR spectra of SF and mSF. It can be seen that the SF before modification, there is a strong infrared absorption peak at 1104 cm^−1^. This is within the range of 1000–1260 cm^−1^ and corresponds with the asymmetric stretching vibration of siloxane and the presence of small absorption peak at 1644 cm^−1^ that is clearly assigned to H-O-H bending. In addition, the absorption peak at 3372 cm^−1^ corresponded to the stretching vibration of silanol groups (Si-OH), as characteristic of silica [27,28]. The FTIR spectra of the mSF modified the SF surface by MPTMS as shown in Figure 1. More difference absorption peaks were observed at 810 cm^−1^, 2840 cm^−1^, and 2940 cm^−1^, respectively, when compared with SF. The SF surface treatment by MPTMS can be identified that the peak at 810 cm^−1^ is due to the stretching vibration of the siloxane (Si–O–Si) bonding [7,28]. The peak at 2840 cm^−1^ and 2940 cm^−1^ is attributed to the stretching of –CH_2_ [29]. However, the disappearance of absorption peak at 1644 cm^−1^ and 3372 cm^−1^ is due to MPTMS reaction to the silanol groups on the SF surface. This clearly indicated that MPTMS can be successfully treated onto the SF surface, which was confirmed by comparing the FTIR spectra of SF and mSF.

### 3.2. Cure Characteristics

The cure behavior of SF, mSF, and calcium carbonate, as a rubber filler in EPDM foam insulation, are showed in Figure 2. Maximum torque (M_H_), minimum torque (M_L_), and differential torque (M_H_-M_L_) were observed to be increasing in proportion to increasing mSF content. This may be due to higher mSF in the EPDM matrix, reducing the rubber chain movement and flexibility. Thus, the mSF enhance the stiffness and modulus of rubber compound which is affect to torque value [30]. The optimum cure time (t_c90_) and scorch time (t_s2_) showed a decreasing trend as increasing mSF contents. It is likely that high shear rate contributed to increased heat build-up in the rubber compound, which led to fast scorch and vulcanization time. The influence of surface modification of SF by MPTMS was also studied. It was found that mSF enhanced the maximum torque (M_H_) and the differential torque (M_H_-M_L_) more than SF because it increased the degree of crosslink density in the rubber chain. However, a decrease in the minimum torque (M_L_) occurred, due to MPTMS reduced the SF-SF interaction. The SF consisted of silanol groups, as shown in the FTIR spectra (Figure 1). This phenomenon caused increased the optimum cure time (t_c90_) and scorch time (t_s2_) more than mSF due to higher accelerator adsorption same as silica usage [29]. So, the benefit of surface modification by MPTMS is not only increased crosslink density but also decreased accelerator adsorption. At a same loading (90 phr), the test results revealed that the maximum torque (M_H_), minimum torque (M_L_) and differential torque (M_H_-M_L_) of mSF were higher than those of calcium carbonate. It is believed that this reinforcement is derived from chemical linkage in the rubber chain, leading to enhance torque. Moreover, the optimum cure time (t_c90_) and scorch time (t_s2_) of mSF are faster than for calcium carbonate. This is due to more heat build-up in the rubber compound and results in faster scorch and vulcanization.

### 3.3. Mechanical Properties

The mechanical properties of SF, mSF, and calcium carbonate as a rubber filler in EPDM foam insulation are summarized in Table 2. The mechanical properties of mSF, such as hardness, tensile strength, 100% and 300% modulus, increased as the mSF loading increased. This is due to the higher differential torque or crosslink density and increasing cohesive force at the mSF–EPDM interface contribute. This confirms that the mSF–EPDM interface contributed to higher differential torque or crosslink density and increased cohesive force. Whereas, the elongation at break decreased as the mSF contents were increased, due to increased maximum torque or stiffness. This had the effect of gradually reducing extension ability since the specific gravity significantly in-creased with increased mSF loading. This may be due to fast scorch and contributed to an increase in the cell wall of EPDM foam insulation, resulting in more difficult expansion. The influence of surface modification of silica fume by MPTMS was also investigated. It was found that silane treatment performed as an excellent reinforcement due to promoted properties significantly superior than non-silane treatment at the same loading [26]. As a result, this confirms that silane treatment increased the chemical linkage and enhanced the mechanical properties such as hardness, tensile strength, 100% and 300% modulus. Moreover, the mSF was shown to have better reinforcing qualities than calcium carbonate. This could be due to chemical linkage reinforcement. All of the test results supported the use of the silica fume as an alternative reinforcing filler that might be expected to replace calcium carbonate in the rubber industry.

### 3.4. EPDM Foam Structure and Properties

The cell structure of EPDM foam directly corresponded to the compression set of EPDM foams as shown in Figure 3 and Figure 4. It can be seen that the cell sizes decreased with increased mSF loading (Figure 3). As the mSF loading was increased, the compression set of EPDM foam gradually decreased (Figure 4). This was due to cell size decrement in the foam matrix, which was produced by expansion of gases from chemical reactions. Therefore, this indicated that smaller cell size gave more durability when exposed to compression. The influence of surface modification of SF by MPTMS was also studied. It was found that the compression set of mSF was lower than SF-reinforced EPDM foam, hence the presence of MPTMS. According to Figure 3, the mSF-filled EPDM foam had a primary cell size smaller than SF-filled EPDM foam. This factor is the key to improve permanent deformation that occurs when a foam is compressed to a specific deformation. Moreover, the comparison between mSF and calcium carbonate were investigated. The mSF exhibits better compression set of EPDM foam than calcium carbonate. The good compression set is obtained with mSF as basis. All of the test results exhibited that mSF could be replaced calcium carbonate in rubber industry.

## 4. Conclusions

MPTMS was used as a silane coupling agent and applied for the modification of SF surface. As FTIR spectra of mSF clearly showed that MPTMS can be successfully treated onto the SF surface. The surface treatment by MPTMS improved the rubber’s torque, which is directly related to crosslink density in the rubber chain. Moreover, the presence of MPTMS showed a beneficial effect through reducing the cure time due to reduced accelerator adsorption. Mechanical properties such as hardness, tensile strength, and modulus were enhanced by mSF when compared to SF and calcium carbonate. The improvement in compression set of EPDM foam is caused by a reduction in cell size in the foam matrix. EPDM foam, reinforced by mSF, provided a primary cell size smaller than SF and calcium carbonate, respectively. All the test results support the replacement of calcium carbonate by mSF for EPDM foam insulation.

## Figures and Tables

**Figure 1 polymers-13-02996-f001:**
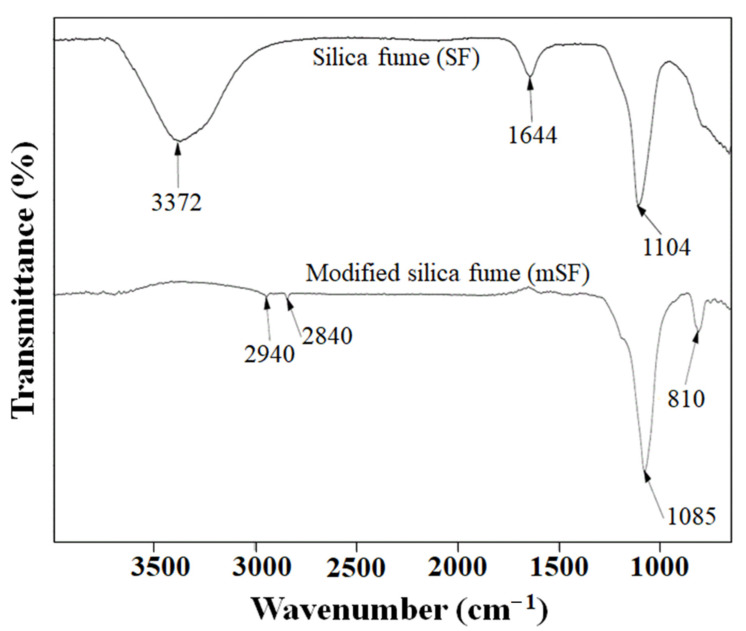
FTIR spectra of silica fume (SF) and modified silica fume (mSF).

**Figure 2 polymers-13-02996-f002:**
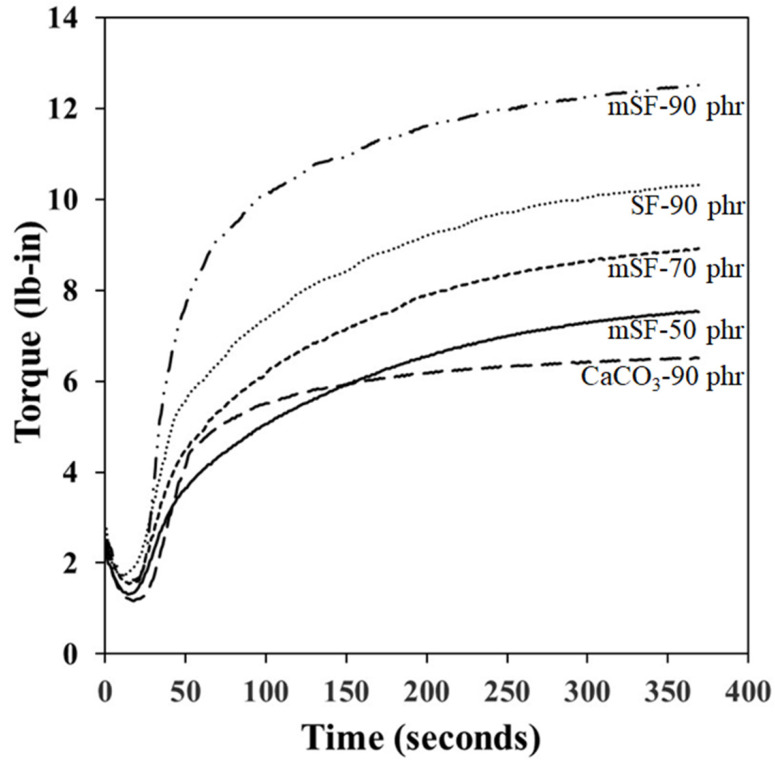
Cure behavior of SF, mSF and calcium carbonate as a rubber filler in EPDM foam insulation.

**Figure 3 polymers-13-02996-f003:**
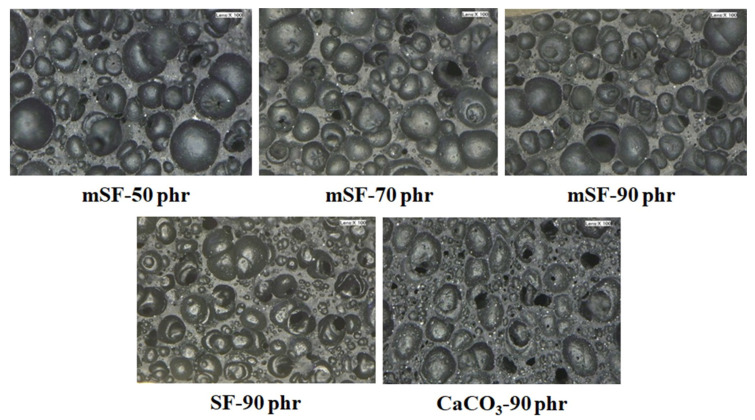
EPDM foam structures using SF, mSF and calcium carbonate as a rubber filler.

**Figure 4 polymers-13-02996-f004:**
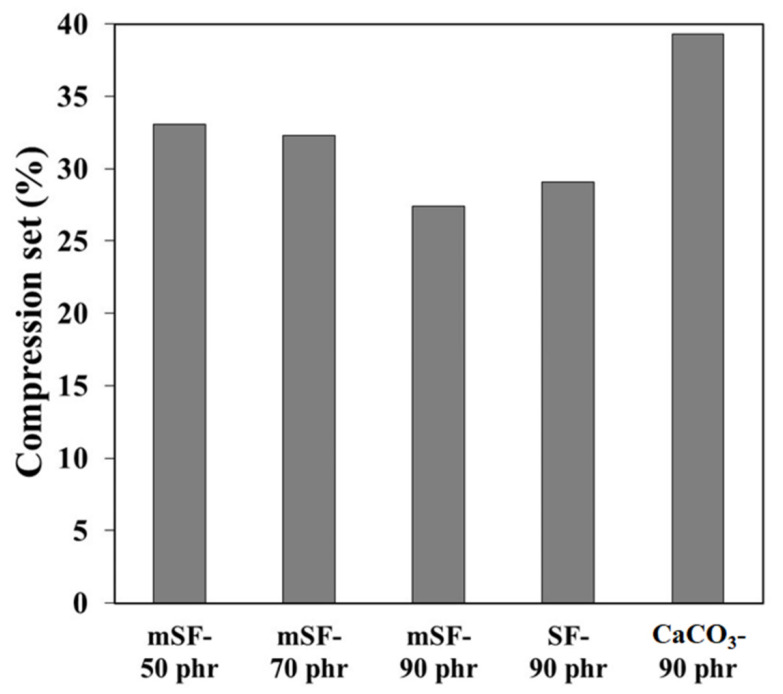
Compression set of EPDM foam.

**Table 1 polymers-13-02996-t001:** Formulations of EPDM foam.

Materials	mSF-50 phr	mSF-70 phr	mSF-90 phr	SF-90 phr	CaCO_3_-90 phr
Keltan 8550C	100	100	100	100	100
N-550	10	10	10	10	10
mSF	50	70	90	-	-
SF	-	-	-	90	-
Omyacarb 1T	-	-	-	-	90
ZnO	5	5	5	5	5
Stearic acid	1	1	1	1	1
PS-430T	5	5	5	5	5
EM-80MA	4	4	4	4	4
TMTD	1.5	1.5	1.5	1.5	1.5
ZDMC	0.5	0.5	0.5	0.5	0.5
MBTS	1	1	1	1	1
Sulphur	1.5	1.5	1.5	1.5	1.5

**Table 2 polymers-13-02996-t002:** Mechanical properties of EPDM foam insulation.

Properties	mSF-50 phr	mSF-70 phr	mSF-90 phr	SF-90 phr	CaCO_3_-90 phr
Hardness (Shore OO)	58 ± 1	62 ± 1	67 ± 1	65 ± 1	55 ± 1
Tensile strength (MPa)	6.26 ± 0.02	6.52 ± 0.01	7.45 ± 0.02	6.62 ± 0.03	5.58 ± 0.02
Elongation at break (%)	520 ± 10	470 ± 15	455 ± 10	460 ± 20	550 ± 10
100% Modulus (MPa)	1.22 ± 0.01	1.51 ± 0.01	1.70 ± 0.01	1.62 ± 0.01	1.18 ± 0.01
300% Modulus (MPa)	3.40 ± 0.02	3.98 ± 0.03	5.10 ± 0.03	4.98 ± 0.01	3.28 ± 0.01
Specific gravity	0.51 ± 0.01	0.54 ± 0.01	0.66 ± 0.01	0.64 ± 0.01	0.45 ± 0.01

## Data Availability

The data presented in this study are available on request from the corresponding author.

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
