# Peer review of "Effect of Modified Silica Fume Using MPTMS for the Enhanced EPDM Foam Insulation"

_polymers, 2021, doi:10.3390/polym13172996_

Round 1

Reviewer 1 Report

This paper used MPTMS to modify the silica fume, then enhanced the EPDM. It is a good job but lacks innovation. Moreover, there are a few suggestion:

  1. In the title, the word “using” should be “Using”, it is a terrible mistake for a research! The whole paper should be revised carefully, there are so many inappropriate expressions.
  2. ALL references are too old, some new paper published in MDPI should be cited.

Reviewer 2 Report

It is recommended to accept after minor revision.

  1. Too many keywords. It should be reduced to less than five phrases.
  2. Different content between mSF-70 phr and mSF-90 phr. But there is no significant difference in elongation at break or even within the error range. Please add a reasonable explanation in the manuscript.
  3. The mechanical properties of mSF-90 phr have not been significantly improved compared to SF-90 phr. Please add to explain whether the design of the SF modification scheme is reasonable.
  4. Please supplement the statistical information of cell size in Figure 3, instead of simply stating the appearance of cell size changes.
  5. The compression set in Figure 4 shows that as the cell area decreases, the compression set decreases, but for SF-90 phr and CaCO3-90 phr, how does the compression set increase as the cell area decreases? Please add a reasonable explanation in the manuscript.